# Diagnostics and Monitoring to Preserve a Hypogeum Site: The Case of the Mithraeum of Marino Laziale (Rome)

**Loredana Luvidi** [1,*]**, Fernanda Prestileo** [2] **, Michela De Paoli** [3]**, Cristiano Riminesi** [4]**, Rachele Manganelli Del Fà** [4] **, Donata Magrini** [4] **and Fabio Fratini** [4]

1 Institute of Heritage Science (CNR-ISPC), National Research Council of Italy, Area della Ricerca di Roma 1, 00010 Rome, Italy
2 Institute of Atmospheric Sciences and Climate (CNR-ISAC), National Research Council of Italy, Area della Ricerca di Roma 2, 00133 Rome, Italy; fernanda.prestileo@cnr.it
3 Conservation Scientist Freelance Marino Laziale (Rome), 00047 Rome, Italy; michela.depaoli82@gmail.com
4 Institute of Heritage Science (CNR-ISPC), National Research Council of Italy, Area della Ricerca di Firenze, 50019 Florence, Italy; cristiano.riminesi@cnr.it (C.R.); rachele.manganellidelfa@cnr.it (R.M.D.F.); donata.magrini@cnr.it (D.M.); fabio.fratini@cnr.it (F.F.)
* Correspondence: loredana.luvidi@cnr.it

**Abstract:** Conservation of hypogea and their accessibility by the visitors is a hard question, due to the interaction of different factors such as the intrinsic characteristics of the hypogeal environments and the presence of public. A particular case is represented by the Mithraeum of Marino Laziale, located a few kilometers away from Rome and accidentally discovered in the 1960s. The uniqueness of the discovery was the presence of a well-preserved painting of the Mithraic scene (II century A.D.) probably due to the oblivion of the place of worship over the centuries as well as the isolation from the outdoor environment. Unfortunately, despite a recent complete restoration and recovery of the archaeological area, which ended in 2015, the area was never open to the visitors and only two years after completing the works it was no longer safe to use. Hence, the need for a new planning of interventions starting from the deep knowledge of this cultural heritage and from the analysis of past incorrect actions to arrive at the opening—without any risk for the archaeological findings and visitors—and management of this site, never exposed to the public. Therefore, since 2018 a diagnostic campaign and microclimate monitoring have been started. The data collected during the two years of investigations have been fundamental to assess the conservation state of the hypogeal environment and the potential risks for the preservation of the three paintings (the Mithraic scene and two dadophores). Long-term monitoring of indoor environmental conditions assumes the role of an essential tool for the planning of preventive conservation strategies but also for the control of the site after its opening to the visitors. Furthermore, the characterization of the microclimate is non-invasive, sufficiently economical and accurate. In this paper, the characterization of surfaces in the Mithraic gallery through optical microscopy, UV fluorescence/imaging techniques, FT-IR spectroscopy, XRD and the microclimatic parameters variation in the presence or absence of visitors are used to define the strategies for the opening and fruition of the Mithraeum. The strategies for the sustainable fruition of this unique archaeological site have been defined through a conservation protocol approved by the Italian Ministry of Cultural Heritage and necessary for the site managers and curators of the Municipality of Marino Laziale to finally support its opening.

**Keywords:** hypogeum; Mithraeum; conservation; frescoes; decay; microclimate; diagnostics

## 1. Introduction

Mithraism was the last of the oriental worships to be introduced into the Roman religion during the Empire. Roman Mithraism gives the god Mithras a very different character from the original one, with the mystical, mysterious and soteriological aspect prevailing. The greatest evidence of the flourishing of oriental worships in Rome can be

found in the III century A.D., especially with the emperor Commodus, who was initiated into the mysteries, followed by Roman high society. The enormous diffusion of Mithraism throughout the Empire was due on the one hand to the favor given to it by the emperors because it facilitated the transfer of the prerogatives of divine sovereignty in their person and was accessible to the spirit of the legions, and on the other hand to the possibility of co-existence with the other divinities of the Empire. Mithraism, being one of the most popular religions among the lower classes, became the most dangerous adversary of Christianity, as it also shared some rituals with it. With the conversion of Emperor Constantine, the decline of Mithraism began until it was completely abandoned in favor of Christianism [1].

Many Mithraea have been found in centers in the East, Egypt, the Balkan peninsula, the Danube countries, England, Germany, France, Spain and Italy, with the worship also spreading thanks to trade relations. Mithraea spread throughout these territories, flanking the temples dedicated to the Greek-Roman divinities, proof that the religion was widespread among all citizens.

Mithraea originated as a type of underground rock architecture and, even when they are built above ground, they always remain strongly linked to this theme.

Originally, the Mithraeum is located in an east-facing cave near a spring. In the absence of a natural cave, the Mithraeum was either built artificially underground or constructed partially or completely above ground and simulated with the asperities of the interior linings.

In their last phase of construction, Mithraea were built inside existing buildings, adapting the rooms to imitate caves. The architectural typology of the Mithraeum reflects the liturgical requirements of the mystery worship: a main hall with a low barrel vault, imitating Mithras' birthplace and alluding to the heavenly vault, with a central corridor between side podiums for mystical banquets and at the end of the *sacellum* the sacred icon depicting the god Mithras [2].

Most of Mithraea were destroyed in the IV century A.D. and those known to us today represent only a minimal part of those that must have been erected. These sites were desecrated and their ritual decorations plundered. This phenomenon has also affected the numerous Mithraea located in the city of Rome and in the Latium Region.

Generally, the situation regarding the state of conservation of Mithraic worship sites is extremely heterogeneous. One of the main problems concerns the coexistence of later stratified structures over time and the hypogeal structures below. Usually, the most well-preserved structures are the hypogean or semi-underground structures. In fact, their burying has protected them from anthropic destruction and the action of atmospheric agents, in the absence of a continuity of use [3,4].

In many of these structures, the original vaults and finishing surfaces are still preserved. However, in the hypogea, the wall paintings, if they have survived anthropic destruction, have suffered from the presence of moisture in the parts in contact with the ground [5]. Water condensation on surfaces can be a major source of damage. In fact, water dissolves salts within the pores of the wall material, and then carries the soluble salts to the surface where, due to evaporation, the salt crystallizes. The result is an inevitable mechanical stress in the painted surfaces. Studies on underground sites have identified as the main causes of the mechanisms of degradation: the chemical processes due to the crystallization of carbonates, which can also take the form of thick incrustations with a strong optical impact on painted surfaces [6,7], and the biological processes due to the formation of microorganisms and algae [8] or, as highlighted by many studies in recent years, by cyanobacteria forming phototrophic biofilms [9–11].

In these environments, the temperature and relative humidity values are almost constant—the humidity rate is above 95% and the temperature values are low—and can favor the growth of microorganisms. These microorganisms can use the stone surface as a support for growth, using the mineral components or surface deposits as metabolites necessary for their development.

Moreover, the presence of visitors could represent a further risk compromising the integrity of the environment equilibrium [12–14]. This equilibrium (temperature, humidity, $CO_2$) could be further changed by the required scene set-up for public opening (such as artificial lighting, power grid, etc.). Furthermore, even if the temperature and relative humidity values are far from those considered optimal for the conservation of wall paintings—according to Cultural Heritage standards and Italian Regulations on conservation [15,16], as they are lower (in the case of temperature) or higher (in the case of relative humidity)—in a confined environment and not affected by the presence of visitors these values establish an equilibrium and remain stable without fluctuations.

On the other hand, short-term or high variations in the indoor relative humidity values can accelerate degradation phenomena: A relative humidity of over 80% can facilitate the biodeterioration and the gradual degradation of organic materials. The monitoring of the environmental parameters is therefore fundamental, considering the possibility of keeping them stable or stabilized over time. In addition, air velocity and, therefore, proper circulation help to maintain the uniform conditions established in the environment, avoiding areas both stagnant and with draft [17].

For this reason, in order to open a hypogeum site to visitors, it is necessary to characterize the environment that has remained undisturbed over time. Once the environment has been characterized, it is possible to proceed with the evaluation of the impact of the public on the internal microclimate [18–20] and on the development of microorganisms active in biodeterioration [21–25]. In fact, the opening of the site to an audience of visitors must be carefully planned considering the influence this will inevitably have on the equilibrium reached for a very long time.

The Mithraeum of Marino Laziale (not far from Rome, in the area known as *Castelli Romani*) is one of these cases, in which the decorations (wall paintings and marble elements) have survived anthropogenic destruction but have been affected by problems related to high humidity and water infiltrations.

In 1962, during the works for the realization of a cellar under a building for residential use, in Marino Laziale, Via Borgo Stazione 12, a long corridor that ended with a painted wall representing the god Mithras was accidentally discovered [4,26,27] (Figure 1). Since its discovery, this exceptional archaeological site has never been officially opened to the public. In the following sixty years, only scholars and sporadic visitors were allowed to visit the site for admiring the archaeological remains and painted scenes, by making an explicit request to the Municipality of Marino Laziale. Finally, in mid-2010, the site was restored, involving both the restoration of three frescoes it holds and the requalification of the area to welcome the public, in front of the Mithraic gallery, the so-called exhibition area. Unfortunately, since the conclusion of these interventions in 2015, the Mithraeum has never been open to visitors. Thus, after only two years of "non-opening", lack of inspection and maintenance, the new exhibition area was in a precarious state of conservation and unsafe for public visits.

Therefore, as will be presented in this study, this site represents an exemplary case of how a restoration work is not enough for its preservation, but a proper conservation activity, including the fruition and maintenance plan should be defined to achieve this goal. The combination of all these actions represents a longer-term vision as defined by the Italian Regulation on the Conservation of Cultural Property and Landscape [28]. Otherwise, without a long-term vision [29–31], at the end of the specific recovery action, everything slowly returns to its initial state and the resources used have been wasted. Use, in a correct way, is a fundamental condition for the preservation of cultural property.

Hence, in order to combine the aspects of conservation with those of promotion and fruition of this hypogeum enriched by the rare pictorial ornamentation, a long-term monitoring of indoor environment parameters (temperature, relative humidity and $CO_2$) and a periodic inspection of the state of conservation of the wall surfaces assume the role of fundamental tools for planning preventive conservation strategies, which are the aim of this study.

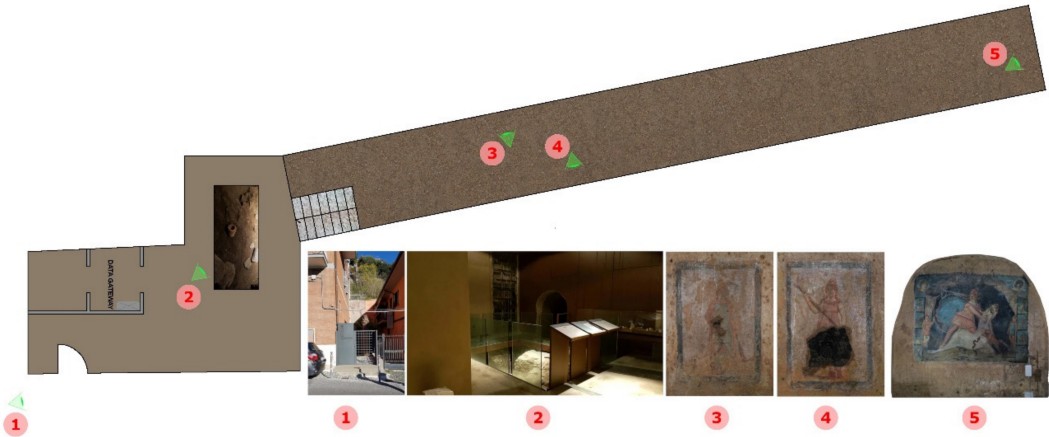

**Figure 1.** Plan of the Mithraeum showing: the entrance on the street (1); the exhibition area (2), with the exhibition set up before 2021; the three painted scenes: *tauroctonia* (5) and the two dadophores, *Cautopates* (3) and *Cautes* (4).

## 2. Case Study: The Mithraeum of Marino Laziale

The place of worship in Marino Laziale was obtained from the barrel-vaulted cistern of a pre-existing Roman villa around the II century A.D. (29 m long, 3.1 m wide and 3 m high). This is pointed out by the presence of the coating plaster waterproofing and curbs at the corners. Probably the religious community that transformed the site into a Mithraic place of worship was constituted by the men who worked in the nearby *peperino* quarries. It is also plausible a second hypothesis that the devotees were the soldiers of the *II Legione Partica*, stationed in the nearby *Castra Albana* (Albano Laziale). In fact, as previously stated, it is known that the Mithraic worship was a religion followed mainly by the less well-off social classes such as soldiers, slaves and workers [26].

The level of the ancient floor of the Mithraeum was about one meter below the floor of the modern cellar, today converted in the exhibition area at the entrance with the purpose of welcoming visitors, and the difference in level was filled by some steps. The transformation of the cistern into a Mithraeum consisted in the realization of a pictorial cycle, representing two dadophores, *Cautes* and *Cautopates,* the helpers of Mithras, on the two sides of the access corridor, and the *tauroctonia*, that is the ritual sacrifice of the white bull by Mithras, with scenes related to the life of the god on the sides, on the end wall (Figure 1) [26].

In the IV century A.D. Christianism spread, becoming the religion of the State and trying to suppress every previous worship; it is likely therefore to think that were just the worshipers of the Mithraeum to close it and to lose its memory in the attempt to save it from assured destruction. The peculiarity that makes the Mithraeum of Marino Laziale an important finding consists both in the existence of a very small number of such worship buildings decorated with wall paintings and in the remarkable state of preservation of the pictorial cycle, which can be traced back to the several centuries of inaccessibility to the site that had determined its oblivion. In fact, if there are many representations of the god Mithras that sacrifices the bull on bas-reliefs, bronzes and terracottas, very rare are instead the pictorial representations on this subject.

In Italy there are only three of these cases: in Rome, under the building of *Palazzo Barberini*; in Santa Maria Capua Vetere (in the province of Naples) and in Marino Laziale indeed [26] despite the fact that in Rome Mithraea were widely distributed throughout the city. In fact it has been assumed that there were around two thousand only in the Rome territory [3].

*Analysis of Past Interventions*

In 2018, before the diagnostic campaign started, archival documents in the property of the Municipality of Marino Laziale were consulted [32]. Through the consultation of archive documents relating to the history of the Mithraeum from the date of its discovery

to the present time, it was possible to retrace all the historical and conservative events that have affected it. The most significant for conservation purposes are briefly listed below. This overview shows that the preservation of this archaeological site has been very complicated over the last decades, both for administrative and conservation issues.

The discovery of the Mithraeum of Marino Laziale was initially kept hidden and the room at the ground of a modern private residential building was used as a cellar until 1963 when the finding was officially communicated to the local Archaeological Superintendence of the Italian Ministry of Cultural Heritage. After a first inspection, it was seen how the improper use of the premises had irreparably compromised the stratigraphy of the site. After ten years, in 1973, the Mithraeum was finally declared Italian state property [26]. However, it was necessary to consider how to solve the problem of access of skilled operators and visitors to the precious monument, making it independent of the above private building.

The first restoration project was presented in 1995, about thirty years after the discovery, with the patronage of INA *Banca di Marino* and supported by a scientific study carried out by the Central Institute of Restoration (ICR) of the Italian Ministry for Cultural Heritage. The project has never been realized.

In 2002, a new inspection by Ministry officials highlighted the need for urgent restoration work, as well as the installation of stationary monitoring equipment to protect the wall paintings. The site was compromised by years of abandonment and inappropriate use. Plaster and paintings were covered by filiform filaments the cause of which had to be investigated.

In 2006, a second restoration project was presented. Microclimatic investigations began for the identification of environmental parameters useful for conservation, as well as a diagnostic campaign under the direction of the ICR for the characterization of the constituent materials of work of art and its phenomena of decay. At that time, the following investigations were carried out by photographic documentation and state of conservation survey, microclimatic monitoring (T, RH, and $CO_2$ measurements), aerobiological measurements, colorimetric measurements on painting as well as chemical, mineralogical and biological analyses and cleaning tests. Following these studies, the restoration project drawn up in 2006 was changed, and definitely approved in 2010 with a public financial contribution by *Provincia di Roma*.

The restoration works started in July 2010 and finished in September 2015. These works consisted in the requalification of the space in front of the Mithraic gallery as entrance room and exhibition area by realizing counter walls, showcases for archaeological findings, lighting system and sanitary facilities (Figure 1), as well as in the restoration of the three paintings. Unfortunately, the Mithraeum has never been open to the public and already in 2017, at the beginning of our diagnostic campaign these recent requalification and musealization interventions were showing great signs of deterioration. There was a significant presence of water in the entrance exhibition area just in front of the Mithraic gallery, both in the form of stains on the floor and on the walls and in the form of drops on the ceiling, as well as the stagnation of water under the stairs leading to the Mithraeum. This presence of water suggested not only surface condensation, due to the high relative humidity of the rooms, but also the presence of infiltration from the areas above.

The new Council of the Municipality of Marino Laziale, also in order to address the urgent necessity to open the archaeological site to the public, in December 2017 commissioned a new diagnostic campaign and a microclimatic monitoring to National Research Council of Italy (CNR). These investigations, to be carried out after 10 years from the last one done in 2008 by the ICR, were aimed to understand the mechanisms of decay of wall paintings and improving the healthiness of environments, essential prerequisites for the purpose of their preservation and the subsequent opening to the public [32].

## 3. Material and Methods

A preliminary inspection of the site was carried out before starting the diagnostic campaign, in order to plan it correctly at every step. A survey of the decay of the walls and the vault of the Mithraic gallery was carried out on the basis of the lexicon of alterations standard and supported by photographic documentation [33,34].

The most appropriate analytical techniques were chosen on the basis of the critical points to be investigated. The first step of the diagnostic campaign is based on the use of non-invasive techniques in situ, in particular imaging techniques (UV–Vis imaging and IR thermography) to assess the state of decay. The measurements were performed on the paintings of the Mithraic gallery and on its lateral walls, in the area surrounding the access stairs and in the environment in front of the access to the gallery used as entrance and exhibition area. These preliminary results were used for focusing both the second step of non-destructive spot analysis, performed by evanescent field dielectrometry (EFD) and adenosine triphosphate (ATP) test, and for specific sampling for some invasive investigations as presented below.

The survey was based on the preliminary documentation by ultraviolet-induced luminescence (UVL) to highlight and locate organic materials eventually present and by a portable digital microscope with different magnification. The combined use of the EFD system and IRT, both on the painted plasters and on the walls of the gallery, made it possible carried out the first mapping of the presence of humidity in the surfaces and to relate it to the possible presence of dissolved salts that could cause alteration phenomena.

Biological analyses were carried out on site through the measurement of ATP, with a bioluminometer, to quantify the possible contamination of surfaces by microorganisms.

Based on the preliminary results of the first analytical steps and addressed by imaging survey, few areas were selected and samples were taken from the surfaces by the use of a lancet, gently scratching the surface to obtain the powder. The sampling was performed following the methodology of Cultural Heritage standard [35]. Sampling was done to gain chemical and morphological information about alteration materials on painted surfaces not otherwise obtainable by non-invasive methods. In Figure 2a is shown the image with the sampling points, listed in Table 1 with relative labels and the descriptions, while in Figure 2b are highlighted the areas investigated by UVL survey.

The micro-samples were observed and documented at high magnification with a microscope under visible and UV light, then micro-fragments from the samples were selected under the microscope and analyzed by Fourier transform infrared spectroscopy (FT-IR) and X-ray diffraction (XRD).

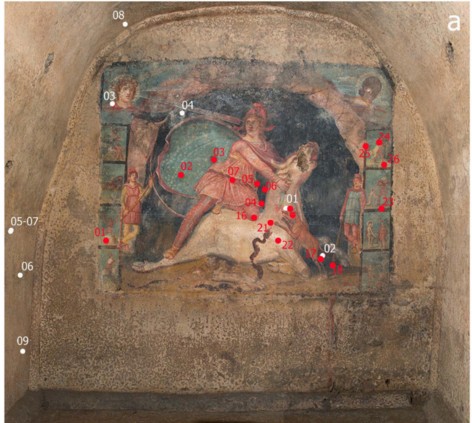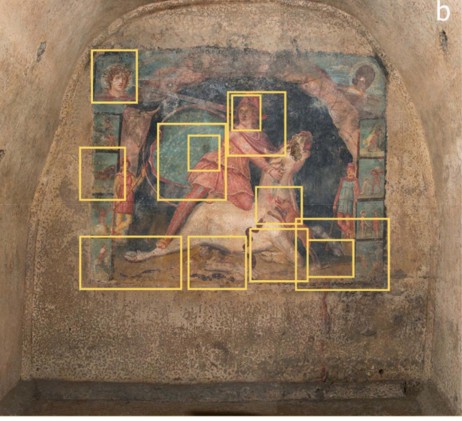

**Figure 2.** Mithraic scene: (**a**) location of sampling points (in white) and microscope images (in red); (**b**) UVL investigations areas.

**Table 1.** Samples analyzed by Fourier transform infrared spectroscopy (FT-IR) and X-ray diffraction (XRD).

| Sample Name | Description of the Sampling Area |
| --- | --- |
| MIT-01 | Mithraic scene: powder from the whitish patina on the dog's snout |
| MIT-02 | Mithraic scene: powder from the white efflorescence under the dog's tail |
| MIT-03 | Mithraic scene: white powder from the head of Medusa in the upper left side |
| MIT-04 | Mithraic scene: powder from the saline efflorescence over Mithra's mantle |
| MIT-05 | Gallery: powder from the black hole in the left side of the wall |
| MIT-06 | Gallery: fragment of *cocciopesto* plaster on the left side, above point "7" |
| MIT-07 | Gallery: shiny black residue in the hole |
| MIT-08 | Dome: powder from the area with efflorescence near the painting |
| MIT-09 | Gallery: clay concretion on the left side of the wall |

### 3.1. Portable Microscope

On the paintings and in the gallery, specific areas were selected and 26 high-magnification images were acquired using a portable microscope. Acquisition of digital images was performed by a Scalar DG-2A portable digital microscope equipped with an optical zoom 25–200×. Images were captured at 25× (area investigated 13 × 8 mm).

### 3.2. Ultraviolet Induced Luminescence (UVL)

To highlight and spatially locate materials with fluorescence emission (such as binders or restoration materials), a documentation by UVL was performed. For the photographic acquisition, a digital camera by Canon EOS 7D (18 Mpixel, CMOS sensor) was used. The camera was equipped with Canon lens EFS 50 mm f/3.5 with a B + W486 UV/IR blocking filter mounted to cut reflected ultraviolet radiation. As sources, two Flash Quantum T5D with B + W UV black 403 filters were used. The same set-up was used for acquiring visible images removing the filter from the flashes [36–39].

### 3.3. Adenosine TriPhosphate (ATP) Test

On surfaces showing luminescence, detected by UVL, the adenosine triphosphate (ATP) measurements were performed to verify the presence of possible active microorganisms using a hand-held ATP bioluminometer (3M[TM] Clean-Trace[TM] NG luminometer, St. Paul, MN, USA). This diagnostic tool is based on ATP bioluminescence technology present in all living cells. The selected areas were swabbed, then the swab was placed inside the bioluminometer and the test activated. The luminometer will measure the light produced: the greater the level of microbial contamination sampled on the swab, the greater the amount of light produced. It is directly proportional to the quantity of living organisms present in the sample. The measure is unable to determine the type of biological contamination.

### 3.4. IR-Thermography (IRT)

The IR imaging investigation on the walls of the exhibition area and paintings was performed by an IR thermal device to investigate the phenomena related to the presence of water infiltration. This instrument detects the temperature of the bodies analyzed by measuring the intensity of the infrared radiation emitted by the surface.

Thermal vision has a wide range of applications: from the detection of humidity to the discovery of hidden architectural elements, from investigation of plaster detachments to the characterization of building materials [40–46]. Using a thermal imaging camera it is possible to perform non-destructive tests that do not require any contact between the equipment and the object under test. The images were acquired by positioning the IR camera frontally on the surface about one-meter distance.

The instrument used to carry out this thermographic survey was a FLIR ThermaCAM B4 (Flir Systems AB, Danderyd, Sweeden) with the following features: spectral range of 7.5 to 13 μm; thermal sensitivity of 0.08 °C (30 °C); thermal image of 320 × 240 pixels;

calibration for building inspections (from −20 to +100 °C) and microbolometric non-cooled FTA detector. The investigations were carried out in passive mode. The images acquired by the thermal camera were later re-processed using the ThermaCAM Quick View software version 2.0, 2006 (Flir Systems AB, Danderyd, Sweeden), in particular to create and align infrared images.

### 3.5. Evanescent Field Dielectrometry (EFD)

The EFD technique is a diagnostic method based on the electromagnetic measurement of the complex permittivity of the material under examination. This physical parameter is specific to the material and describes how the material can modify the electric field E applied to it. This method has been usefully employed to non-invasively determine the moisture and salt content within materials [47,48].

The EFD system called SUSI™ (this trademark is the Italian acronym for integrated instrument for measuring of moisture and salts content [49]) consists of a microwave signal meter, the probe, and a notebook on which the software for controlling the instrument is installed, as well as for real-time data processing. This instrument allows to carry out sub-surface measurements up to 2 cm in depth of the water amount present within the substrate (moisture content), and the salinity index (the index ranges from 1 to 10, where 1 represents the absence of salts and 10 the presence of superficial efflorescence; however, does not allow to distinguish the salts nature).

### 3.6. Fourier Transform Infrared Spectroscopy (FTIR)

FTIR data in attenuated total reflection（ATR）mode were collected on micro-fragments using an ALPHA FT-IR (Bruker Optics). Spectra were recorded in the operating range 4000–400 $cm^{-1}$, with 4 $cm^{-1}$ resolution, and 64 scans. Bruker Opus 7.2 software (Bruker Corporation, Billerica, MA, USA) was used for data acquisition and processing [50–53].

### 3.7. X-ray Diffraction (XRD)

Diffraction patterns were recorded by using a powder X-ray diffractometer PANalytical X'PertPRO equipped with X'Celerator multi revelatory and High Score data acquisition and interpretation software (Cu anticathode ($\lambda$ = 1.54 Å)), under the following conditions: current intensity of 30 mA, voltage 40 kV, explored 2θ range between 3 and 70°, step size 0.033°, time to step 60 s [54].

### 3.8. Microclimate Monitoring

The microclimatic monitoring campaign covered the two areas of the archaeological site: the entrance with archaeological ruins and exhibition area and the hypogeum environment of the Mithraic gallery. The microclimatic monitoring was carried out for a period of two years, from August 2018 to August 2020, in order to analyze the thermo-hygrometric conditions of the site before the start of the last recovery works aimed at opening the Mithraeum to the public.

The indoor monitoring system was positioned according to Cultural Heritage standards [15,16,55,56], as shown in Figure 3, at different distances from the entrance of the gallery up to the wall where the Mithraic scene is depicted and at different heights (50 cm, 150 cm, on the vault) from the floor to the vault. For fixing the sensors in a hypogeum environment, low-impact supports made with a stainless material suitable for an environment with high relative humidity values were adopted.

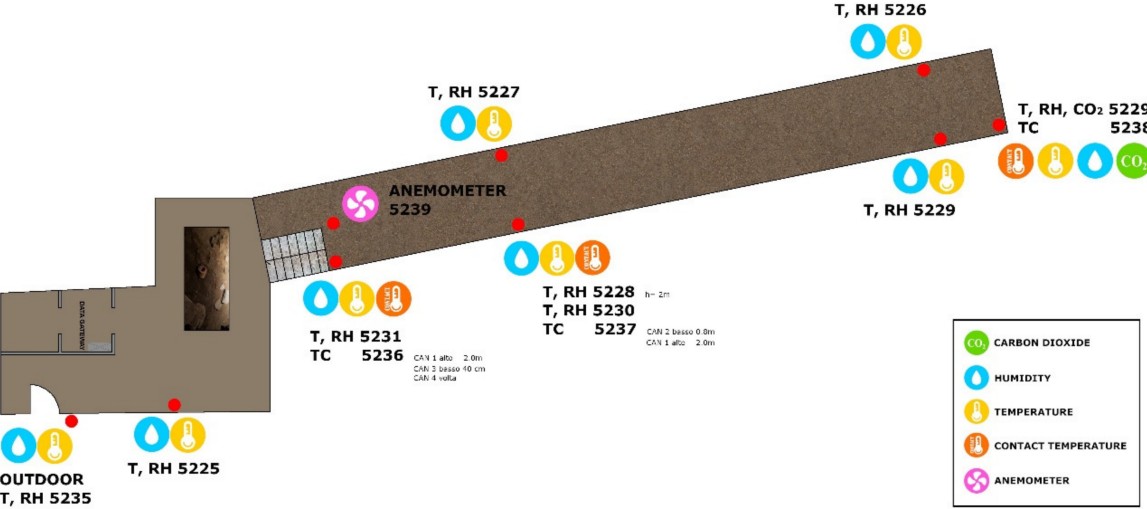

**Figure 3.** Plan of the Mithraeum with the location of the sensors.

At the same time, a sensor of temperature and relative humidity was placed in the entrance/exhibition area that has no openings or air changes with the exception of a small grate overlooking the street. Another sensor was positioned outdoors of the archaeological site, in order to make a comparison with the seasonal trends of these parameters.

Specifically, a system of Delta Ohm sensors were placed to detect the following microclimatic/physical parameters: 8 temperature and relative humidity sensors (HD 35EDW model); 6 surface temperature sensors (HD 35EDW model); 1 $CO_2$ sensor (HD 35EDW model); 1 sensor for measuring air flows (sonic anemometer) (HD52.3D model) (Figure 3).

The combined system of sensors was connected to a single base station with a data acquisition device through Wi-Fi network, then a Global System for Mobile Communications (GSM) transmits to an external cloud the monitoring data in order to be analyzed. All the collected data can be accessed via a dedicated Delta Ohm web interface, through which it is also possible to visualize and download the previously acquired data. Each sensor was configured to acquire data every hour.

## 4. Results and Discussions

### 4.1. Diagnostic Investigations

Various information emerged from the results of the diagnostic campaign carried out and from the microclimatic monitoring. The documentation performed through the portable microscope was addressed to study the conservation state of the wall painting and to document the details of the pictorial decoration.

The documentation was also carried out in correspondence with the sampling areas. As an example, two details are reported relating to the presence of whitish patinas due to salt crystallizations, highlighted by the high magnification images, acquired on the snout (MIT-20) and on the tail (MIT-18) of the dog (Figures 2a and 4a,b).

Moreover, observations at high magnification performed by portable microscope, highlighted peculiar details, such as brushstrokes (MIT-25) and some intentional scratches (MIT-23) (Figure 4c,d).

The graphic survey of the alterations of the surfaces of the gallery does not highlight the critical aspects shown by the survey of 2008 [32], while the effects of the restoration interventions carried out between 2014 and 2015 are still visible.

In fact, during the 2008 diagnostic campaign, thematic tables were drawn up on the constituent materials and the phenomena of alteration present both on the walls of the gallery and on the painting. These tables were updated in 2018 (Figure 5). The recurrence of the whitening of the surfaces due to the concentration of carbonate salts was observed.

In particular, the painted wall presents the same problem spread over the entire pictorial layer but in less concentration than 10 years ago, as only four years have passed since the end of the restoration.

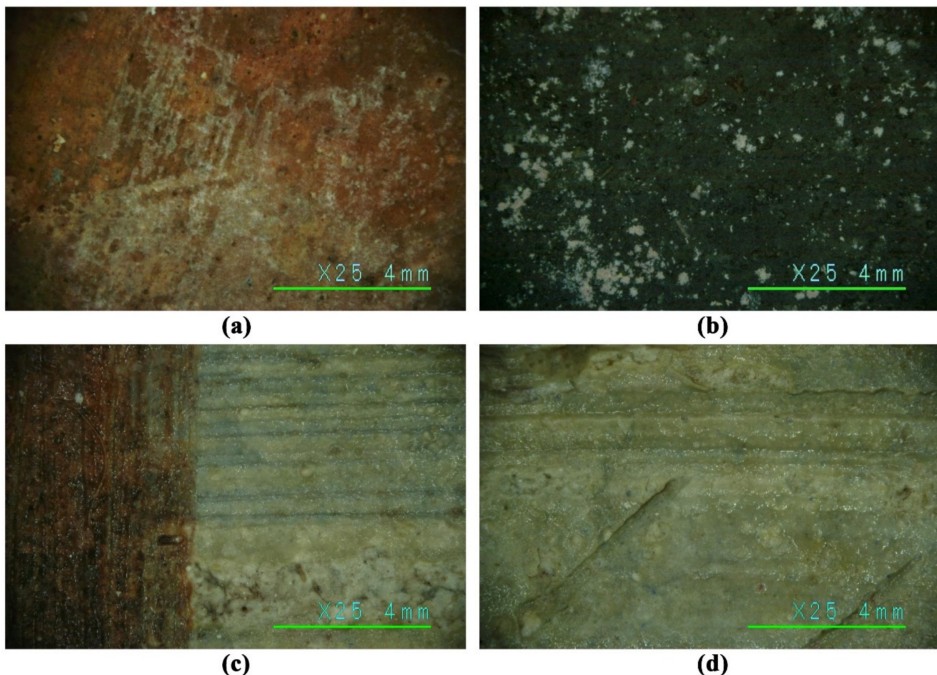

**Figure 4.** Details with high magnification by portable microscope: (**a**) Snout of the dog; (**b**) tail of the dog; (**c**) brushstroke on the first panel at the top from the right side, under the legs of the bull; (**d**) scratch on the first panel at the top from the right side, under Mithra's feet.

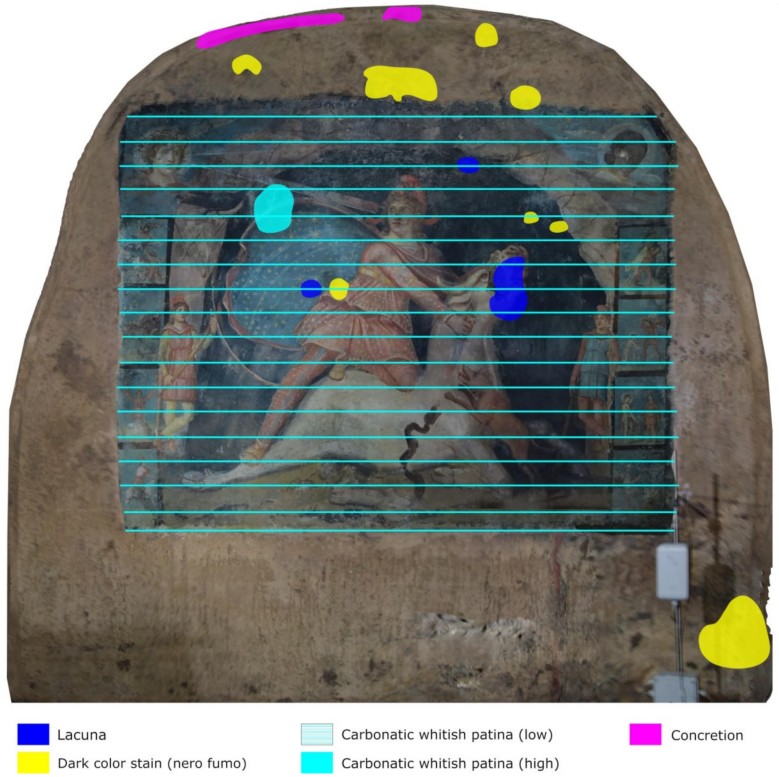

**Figure 5.** Wall with Mithraic scene: map of alteration phenomena (October 2018).

The UVL survey carried out on the three painted scenes highlights some areas affected by a light-bluish fluorescence response due to the presence of salt efflorescence patinas (Figure 6).

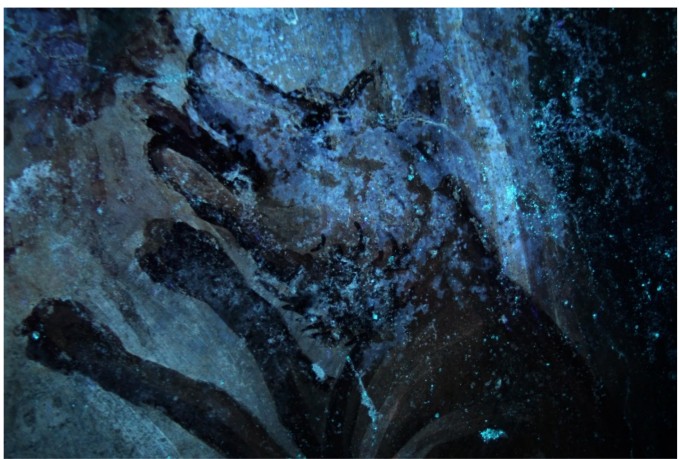

**Figure 6.** Mithraic scene: details of UVL on the snout of the dog.

FT-IR analysis carried out on micro-samples picked up from these whitish patina pointed out the presence of calcium carbonate. This evidence is confirmed by the results obtained by XRD that highlights the ongoing carbonatic crystallization. In some samples, the carbonatic precipitation incorporates silicatic impurities, probably coming from the substrate on which the tunnel was excavated.

In detail, the spectra of the samples show the same absorption bands at 1408, 872 and 712 cm$^{-1}$ assignable to calcite [57–59]. The absence of high frequency bands suggests that the mineral is not deteriorated (powdery). Absorption bands at 1014 and 453 cm$^{-1}$ suggest the presence of silicates attributable to mica (Figure 7) [59].

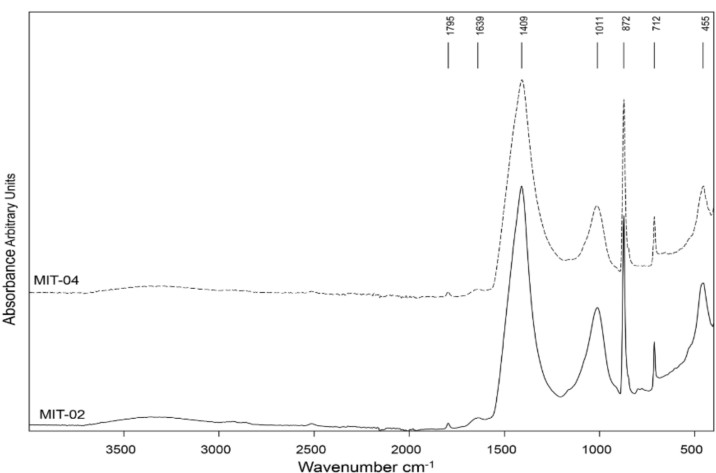

**Figure 7.** ATR FT-IR spectra of MIT-02 (line) and MIT-04 (dots) samples.

This result is confirmed by XRD analyses, which identify the annite mineral. This mineral is linked to the composition of the stone substrate brought to the surface by capillary rising.

In a few areas, a yellowish fluorescence emission was documented. This phenomenon is probably due to the presence of organic substances. This organic material is likely to have been applied to the painting as a protective layer during the 2014 conservation treatment, not specified. This material looks shiny and spread with thick and full-bodied brushstrokes.

It was currently not characterized by the authors as it was detected in central areas of the Mithraic scene and, therefore, it was not possible to carry out sampling that was not strictly necessary for the conservation purposes of the painting.

EFD analyses were carried out on the Mithraic scene, in a similar way as in Di Tullio et al. [60], in order to evaluate the moisture content (MC) and the salinity index (SI).

The amount of MC is higher in the left area of the painting and it decreases from the center to the right side of the wall (Figure 8a). On the contrary, the salinity index SI shows an opposite situation: lower values on the left side and higher ones from the center to the right side (Figure 8b).

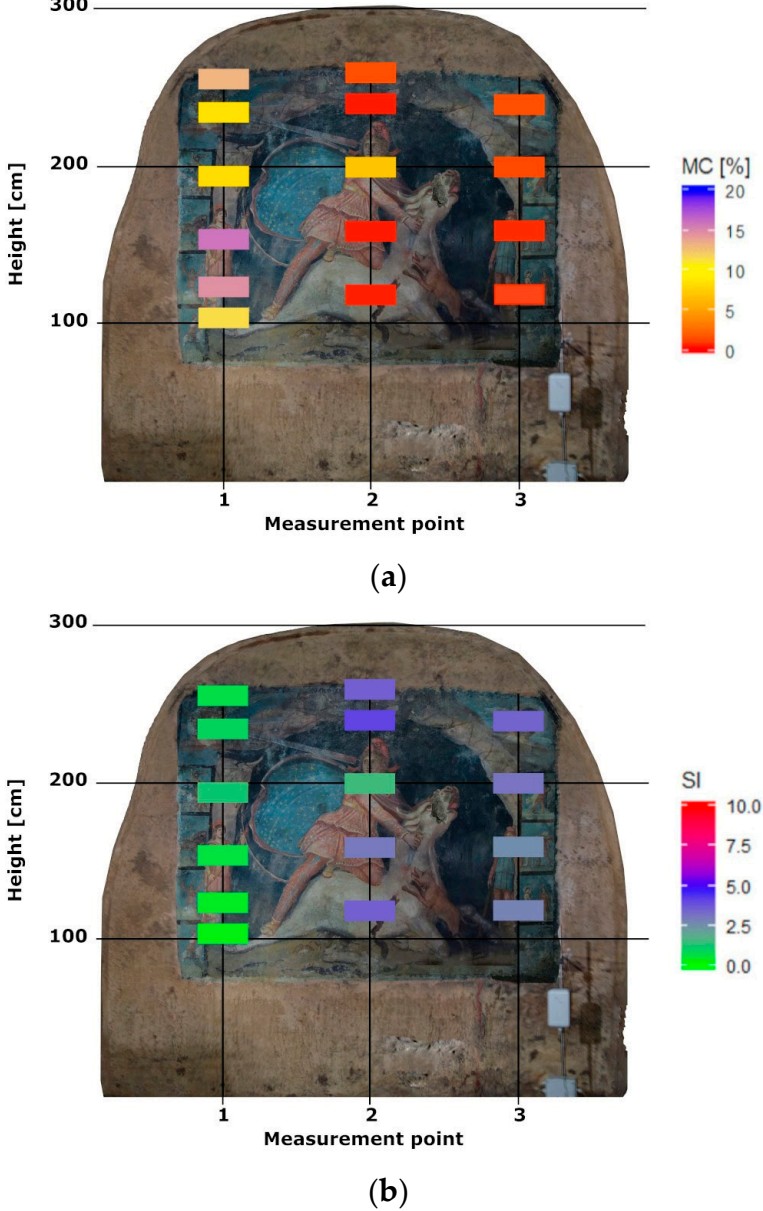

**Figure 8.** Mithraic scene, measurements by EDS on the surface of the painting: (**a**) distribution of moisture content (MC) and (**b**) salinity index (SI).

This phenomenon could be attributed to the fact that, in the areas where the moisture content shows higher values, the salts are more diluted and, therefore, the SI salinity index is lower.

EFD analyses were carried out also on the wall under the stairs at the beginning of the Mithraic gallery, where the most water accumulates. In this area, the MC is higher than on

the Mithraic scene. On the contrary, the concentration values of salts (SI) are lower in the presence of high percentages of humidity, probably due to the fact that in the wettest areas the salts are more diluted.

The IRT investigations, in passive mode, were carried out for an initial mapping of the areas with greater degradation due to water (infiltration and rising damp) and have provided significant results only in the exhibition room in front of the Mithraic gallery; in this area, where the counter-walls were removed, the investigation highlighted an important presence of humidity.

This phenomenon has determined the degradation of the metal structures used to support the counter-walls. The oxidative process of the metal elements has generated chromatic variations into reddish color on the counter walls, strongly altering their original aesthetic appearance. Therefore, in the requalification works, the counter-walls were detached from the perimeter walls.

On the contrary, on the main painting in the Mithraic gallery and on the two small dadophores painted in the two sides of the gallery, did not highlight any particular critical issues.

ATP measurements, with RLU (relative light unit) values ranging between 4000 and 12,000, indicated the presence of metabolically active microbes on the Mithraic scene and also in some places on the rest of the gallery. The important biological contamination is due to the presence of insects on the walls gallery and from the food cycle of these animals. It is also consistent with the environmental conditions of high humidity and high water content on the surfaces.

However, on the basis of the graphic survey on the state of decay of the surfaces and the diagnostic campaign results, it is evident that this biological contamination does not produce biodeterioration. This situation is in agreement with the results of the microbiological investigations carried out in 2007 by the Central Institute of Restoration of the Italian Ministry of Cultural Heritage [32].

In addition, since the Mithraeum is closed to the public, it has not yet been necessary to evaluate the risk of biological contamination induced by the presence of visitors. This kind of evaluation must be taken into account when planning the action of opening the site, monitoring the situation over the time.

*4.2. Microclimate Monitoring*

Until May 2019 the Mithraeum was closed to the public for safety reasons, but few visits by scholars or professionals were allowed only upon explicit request to the Municipality of Marino Laziale. After this date, the Mithraeum was totally closed to allow new recovery works to be carried out in the exhibition area in front of the Mithraic gallery. The works started in May 2019 and finished in June 2019 allowed the reopening of the two existing windows and a first inspection on the origin of the water flows.

When the diagnostic campaign began in June 2018, the LED spot lighting system installed along the Mithraic Gallery during the 2015 restoration was already out of order. After two years of microclimatic monitoring, from August 2018 to August 2020, carried out in the exhibition area and inside the Mithraic gallery, in comparison with the external conditions, a situation of substantial stability and uniformity of the temperature and relative humidity values emerged. Specifically, in summer before the reopening of the two original windows, the temperature values are between 20 and 21 °C in the entrance/exhibition area and 16–17 °C in the gallery. Any significant differences were measured either at different lengths and heights of the Mithraic gallery (Appendix A—Figures A1 and A2). The surface temperature sensors revealed an increase of 1–2 degrees starting from the entrance and going inwards the gallery and of 2–3 degrees from the floor level up to the bottom level, ranging from 14 to 17 °C. In winter 2018/2019, the air temperature of the two zones (Mithraic gallery and exhibition area) varies from 12 to 14 °C (Appendix A—Figures A3 and A4) while the surface temperature of the gallery ranges between 9 and 14 °C.

Inside the gallery, the relative humidity values are stable (summer and winter around 100%) while in the exhibition area are revealed slight oscillations with values between 94% and 97% during the winter (Figures 9 and 10). In summer, similar values have been measured, until the end of June 2019 when the reopening of the windows was completed.

Instead, after the reopening of the two windows, the temperature values in summer are between 17 and 23 °C in the exhibition area and 14–18 °C in the Mithraic gallery. Even in this period, different values of temperature and contact temperature were measured either at different lengths and heights of the Mithraic gallery. In the winter 2019/2020, the air temperature for both areas varies from 10 to 14 °C, while the surface temperature ranges between 9 and 14 °C.

Inside the Mithraic gallery, both in summer and in winter during the whole period of monitoring, the values of relative humidity are around 100%; therefore, stable also after the recovery works.

On the other hand, in the exhibition area the relative humidity values range from 75% to 100%, with a maximum reduction registered compared to the Mithraic gallery of 25% in summer (Figures 11 and 12). This variation is probably due to a greater influx of hot air into the exhibition area in the summer/autumn period following the reopening of the two windows. However, the environmental conditions of the exhibition area do not affect the thermo-hygrometric behavior of the Mithraic gallery.

There are no air flows in the gallery measured by the sensor during the different seasons. Negligible values were also measured during the presence of authorized small groups of visitors.

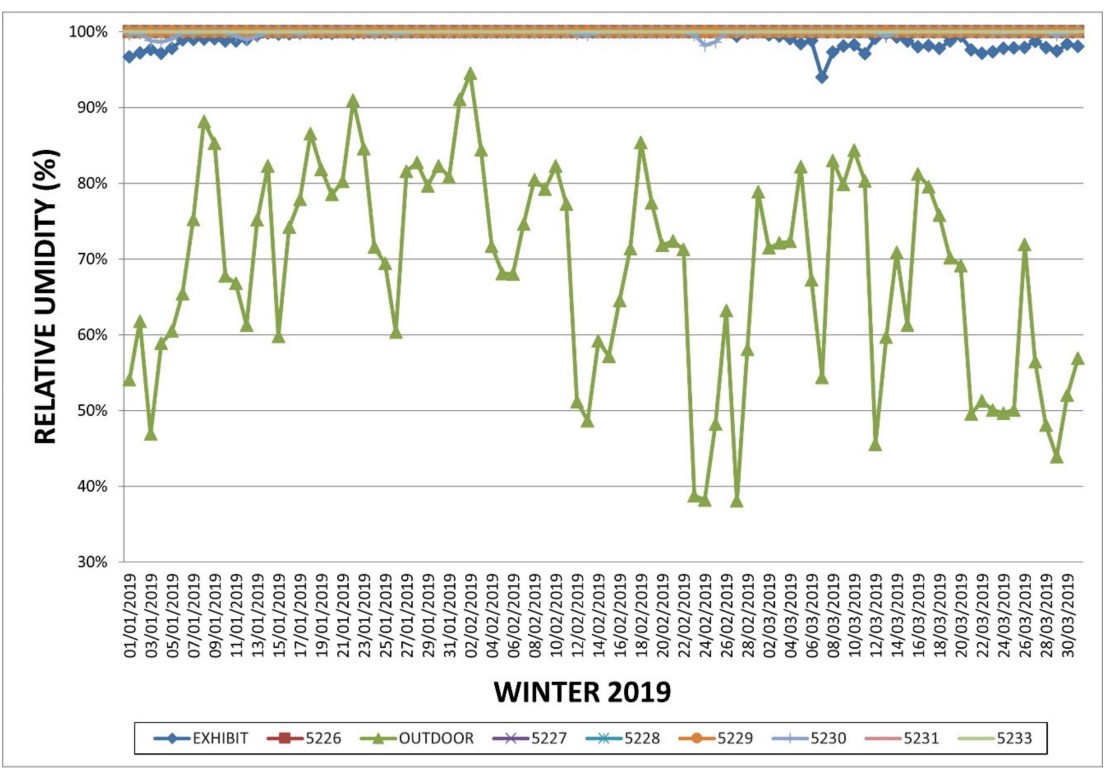

**Figure 9.** Indoor relative humidity values inside the exhibition area and the Mithraic gallery in comparison with the outdoor values during winter 2019 (from 1 January to 31 March).

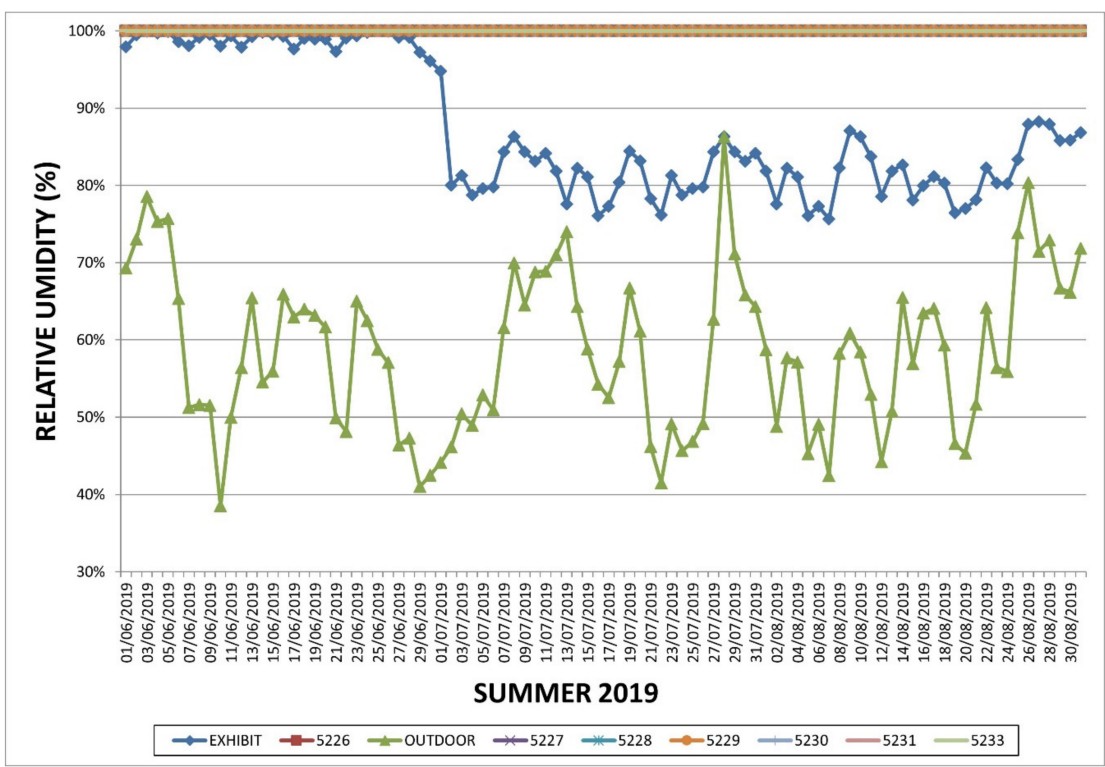

**Figure 10.** Indoor relative humidity values inside the exhibition area and the Mithraic gallery in comparison with the outdoor values during summer 2019 (from 1 June to 31 August).

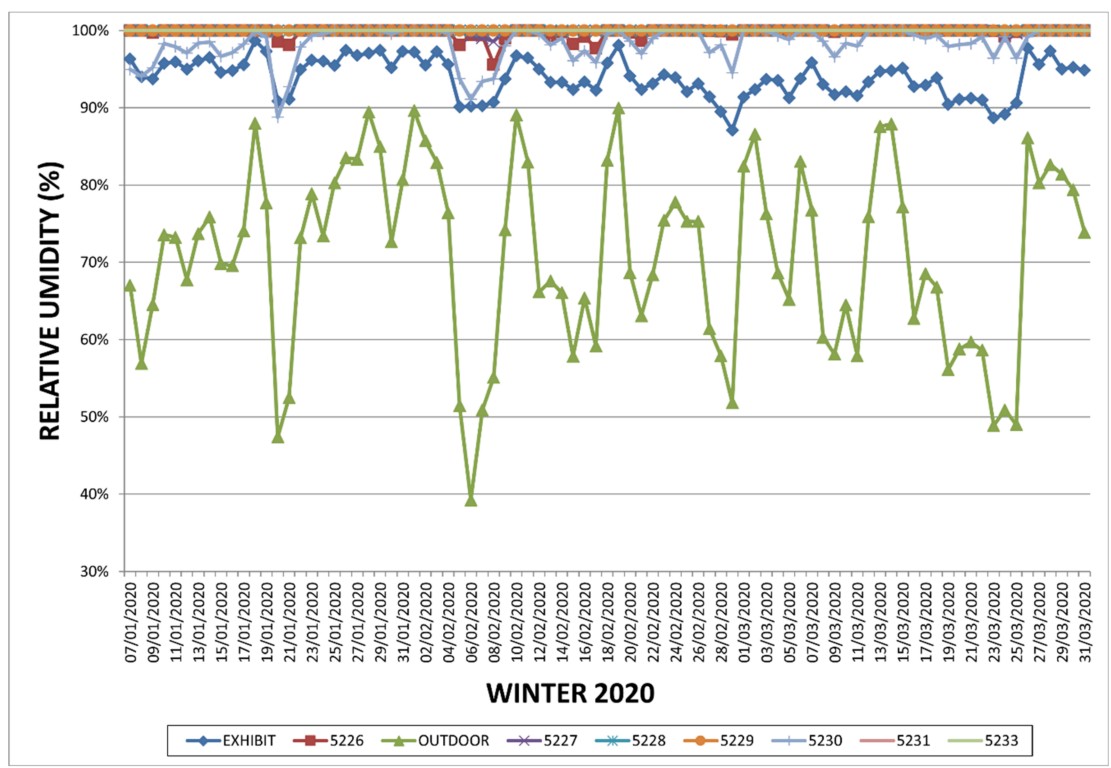

**Figure 11.** Indoor relative humidity values inside the exhibition area and the Mithraic gallery in comparison with the outdoor values during winter 2020 (from 7 January to 31 March).

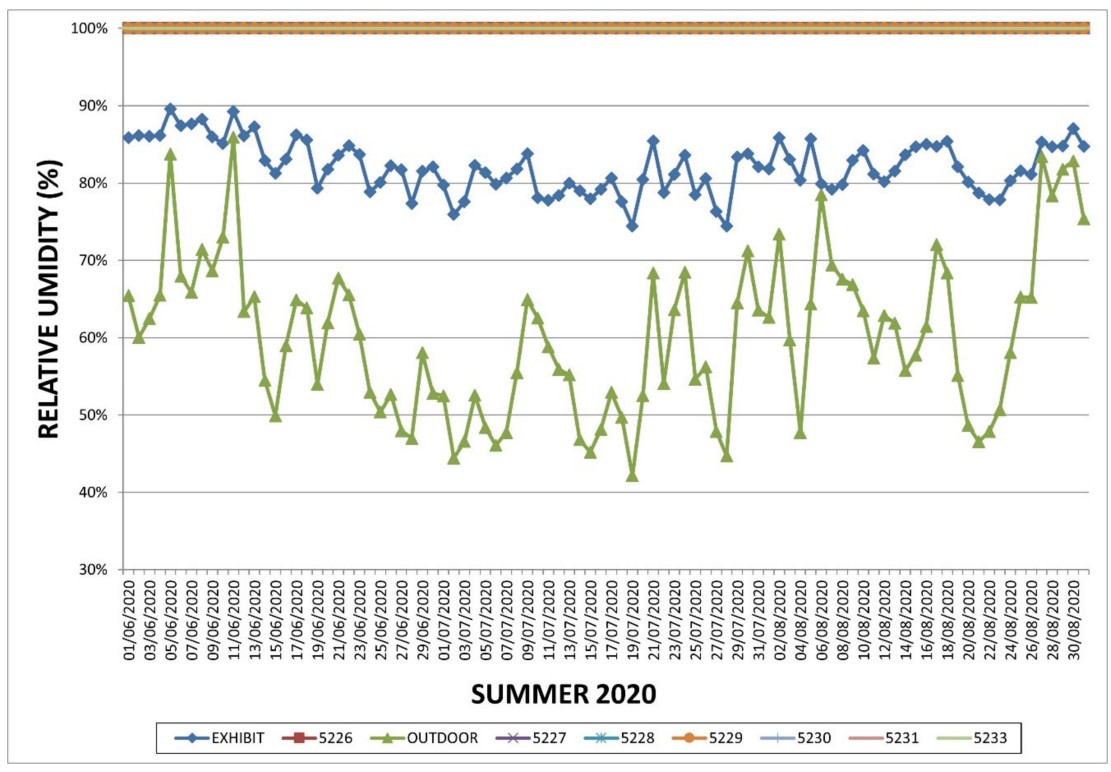

**Figure 12.** Indoor relative humidity values inside the exhibition area and the Mithraic gallery in comparison with the outdoor values during summer 2020 (from 1 June to 31 August).

To summarize, the Mithraic gallery is not influenced by external variations in temperature and relative humidity, and even in the exhibition area these values undergo very small variations. Variations are detectable in the long period in relation to seasonal change but this variation is not highlighted in the daily acquisition. As expected, in the warmer months, the internal temperature is lower than the external one, while in the cold months the opposite situation occurs.

Concerning the $CO_2$ measurements of the Mithraic gallery, in absence of visitors, the measured value is of approximately 500 ppm. In the isolated cases where there was a planned and authorized presence of visitors within the archaeological area, it was found that the values rise to 700–800 ppm when groups of 3–4 people stay in the Mithraic gallery for a couple of hours (a value that immediately drops after an hour), while in the event of a large number of visitors (about 15), the value rises above 1000 ppm and takes several hours to drop back to 500 ppm. A recent study indicates that the first effects on humans begin at concentrations of about 1000 ppm $CO_2$ [61], so the indoor measured values can be considered safe.

The effect of an increase in $CO_2$ with respect to the conservation of the paintings has also been taken into consideration because, according to its abundance, it can cause dissolution and crystallization phenomena of carbonates. In particular, the dissolution is dangerous for the existence of the plaster supporting the painting while the carbonate crystallization on the surfaces, a widespread phenomenon in underground sites, can compromise the legibility of the paintings and modify the surface structure. On the other hand, the time required for recrystallization, due to the dissolution-precipitation reactions initiated by the surface condensation of $CO_2$-rich water, is very long and undetectable in a limited number of years [6,21]. Furthermore, the phenomenon of sudden release of gases, mainly $CO_2$, from the soil must also be considered, since the site is located in the *Colli Albani* volcanic region [62,63].

## 5. Conclusions

In conclusion, from the analysis of the results acquired during the diagnostic campaign and the microclimatic monitoring, it is possible to state that the Mithraic gallery and the paintings inside it are in good state of conservation, after more than ten years since the previous diagnostic campaign (2008) and five years since the restoration of the painted surfaces (2015). In fact, there are not substantial alterations of their state of conservation. The microclimatic monitoring has highlighted a condition of stability of the parameters of temperature and relative humidity, except for slight variations in the exhibition area in front of the gallery Mithraic following the requalification works in 2019. A new microclimatic monitoring campaign is planned after the regular opening of the site to the public, with the specific goal to monitor environmental parameters during scheduled limited visits.

Starting from this state of art, the right of public access to this unique and fragile cultural heritage [64], which has never been visited since its discovery which dates back to the 1960s (except for some scholars, technicians and managers), is considered a necessary process for the "heritage-making", as reported in Duval [65], but following the principle of maximum precaution, visits must be supported by a protocol that make it possible the conservation and transmission of the heritage over time. Therefore, opening to the public requires continuous monitoring of the microclimatic parameters and periodic inspections of the state of conservation (three to four times per year) for the correct configuration of the permitted visit protocol. As stated, the presence of the public can lead to biodeterioration phenomena due to the microbial community introduced by visitors, by variations in the microclimatic parameters T, RH, $CO_2$ and air flows, and attention must also be paid to the effect of the lighting system.

This recent survey in the Mithraeum carried out by the CNR was, therefore, the driving force for designing improvements in the management of this precious and practically inaccessible archaeological site to the community. Particular attention was paid to the state of conservation of the three paintings, the *tauroctonia* and two dadophores, to the environmental conditions of the Mithraic gallery and the exhibition area. In addition, a continuous search for new and more appropriate exhibition solutions, both from the point of view of paintings conservation and safety for visitors, in an underground environment characterized by high relative humidity and problems of water infiltrations where the *cocciopesto* is no longer on the vault, must be carried on. The study of the programmed and contingent accesses, during the microclimatic monitoring period, of small groups of people (from 5 to max 15) for a limited time (15 min in the exhibition area and 15 min inside the gallery) gave preliminary indications on the variations of $CO_2$ and T/RH introduced by human presence as well as air circulations. Unfortunately, the need to make the site safe and the impact of the Covid pandemic on visits to narrow indoor environments has slowed down a more systematic assessment. Following the conclusion of the CNR study and on the basis of the results presented, the Municipality of Marino Laziale, with the approval of the Superintendence (*MIC—Soprintendenza Archeologia, Belle Arti e Paesaggio per l'area Metropolitana di Roma e per la Provincia di Rieti*), commissioned a new project for the recovery of the archaeological site in order to permanently open it to the public. In spite of the difficulties caused by the pandemic situation, the recovery works started in 2021, were carried out and completed, allowing the inauguration and opening to the public in September 2021.

The long and complex history of the conservation of the Mithraeum of Marino Laziale, which has lasted sixty years, from its discovery through the various restoration and exhibition projects, ending with what should have been its first opening to the public in 2015, has clearly demonstrated the need for a long-term vision. Without such a vision, resulting from correct planning and management, at the end of a specific recovery action everything slowly returns to its initial state. In this way, the resources used for the recovery action will have been lost.

Therefore a correct fruition of the site and preventive conservation are fundamental conditions for the preservation of cultural heritage to be passed to future generations.

**Author Contributions:** Conceptualization, L.L., F.P., M.D.P.; methodology, L.L. and F.P.; investigation, L.L., F.P., M.D.P., C.R., D.M., R.M.D.F., F.F.; data curation, L.L., F.P., M.D.P., C.R., D.M., R.M.D.F., F.F.; writing—original draft preparation, L.L., F.P., M.D.P., C.R., D.M., R.M.D.F.; graphical editing, R.M.D.F., M.D.P., D.M.; writing-review and editing, L.L., F.P., M.D.P., C.R., D.M., R.M.D.F.; funding acquisition L.L.; supervision L.L. and F.P. All authors have read and agreed to the published version of the manuscript.

**Funding:** This research was co-financed through the following projects: "Study of the microclimatic conditions within the archaeological area of the Mithraeum and related diagnostic campaign", executive resolution no. 1229 of 20 December 2017 by the Municipality of Marino Laziale (Rome); "Functional and technological requalification interventions of the archaeological area of Dio Mitra sanctuary", executive resolution no. 950 of 12 October 2020 by the Municipality of Marino Laziale (Rome) on funds from the Italian Ministry for Cultural Heritage and Activities—PON "Culture and Development"—FESR 2014–2020 and the PON "Culture and Tourism"—FSC 2014–2020.

**Institutional Review Board Statement:** Not applicable.

**Informed Consent Statement:** Not applicable.

**Data Availability Statement:** Not applicable.

**Acknowledgments:** The authors would like to thank Rosa Anna Senese and Cristina Aufiero students of the University of Roma Tre for the graphic surveys of the conservation state, carried out in October 2018, during their internship at the CNR-ICVBC in Rome, under the supervision of Loredana Luvidi and Fernanda Prestileo. The authors would also like warmly to thank Stella Nunziante Cesaro for assisting us with the FT-IR analysis carried out in the laboratory.

**Conflicts of Interest:** The authors declare no conflict of interest. The funders had no role in the design of the study; in the collection, analyses, or interpretation of data; in the writing of the manuscript, or in the decision to publish the results.

## Appendix A

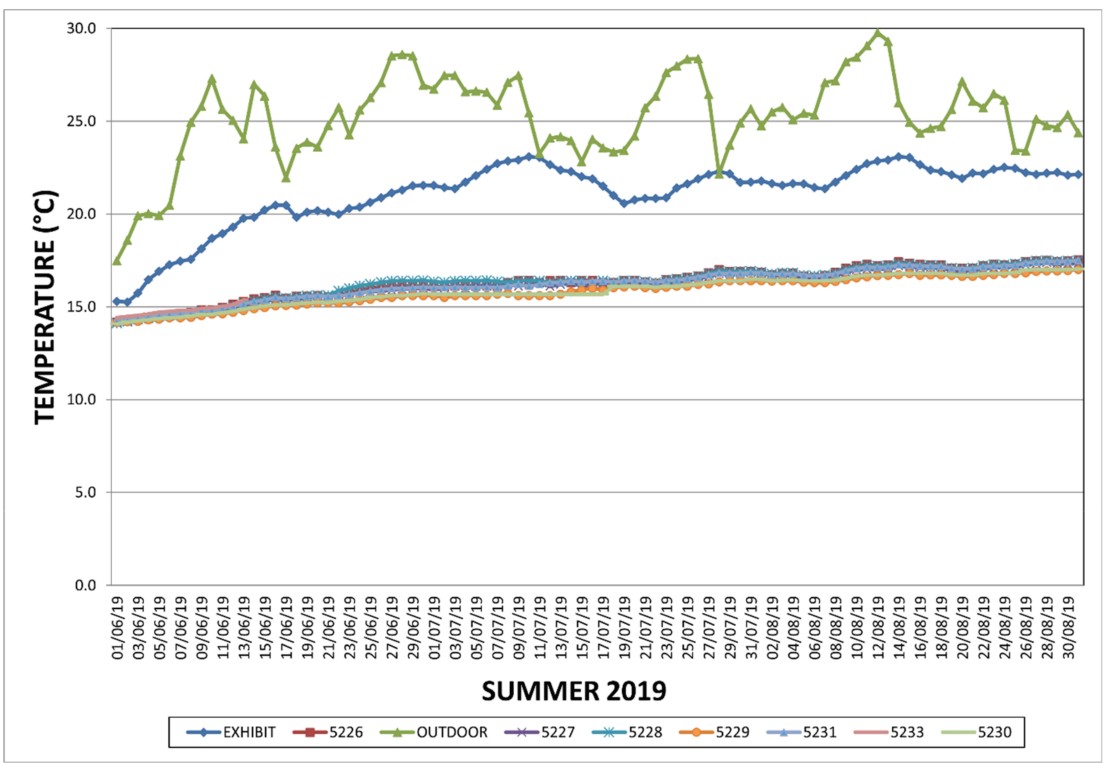

**Figure A1.** The indoor temperature values inside the exhibition area and the Mithraic gallery in comparison with the outdoor values during summer 2019 (from 1 June to 31 August).

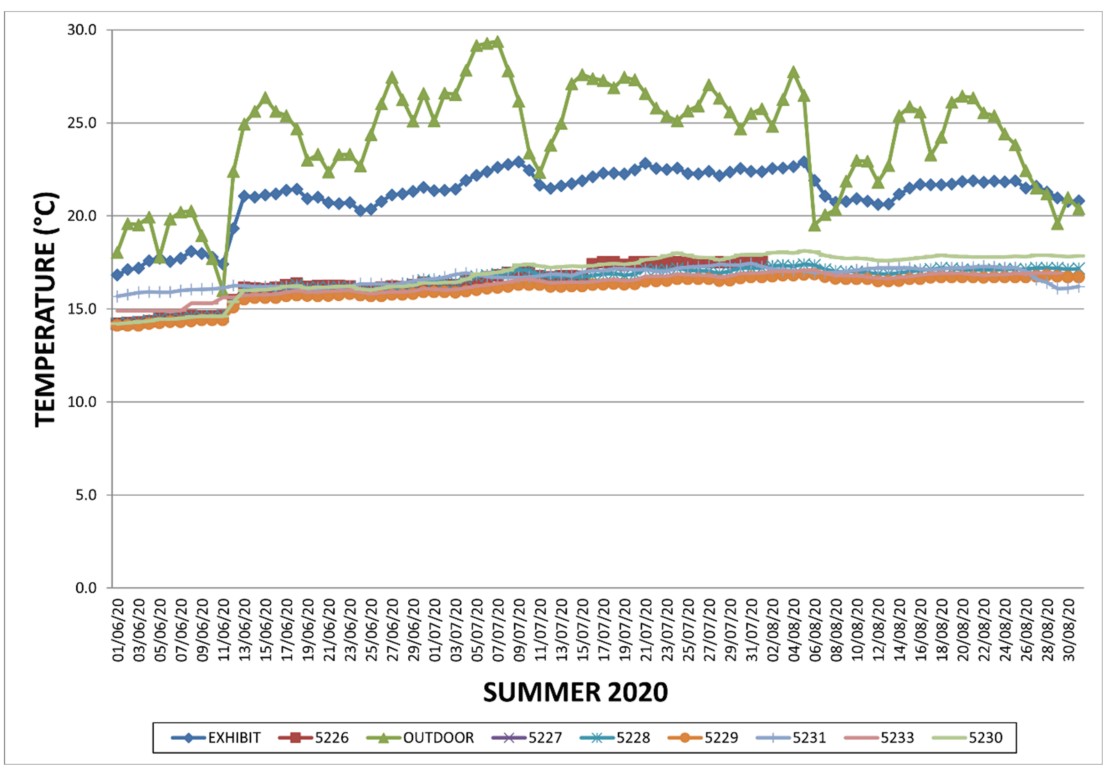

**Figure A2.** The indoor temperature values inside the exhibition area and the Mithraic gallery in comparison with the outdoor values during summer 2020 (from 1 June to 31 August).

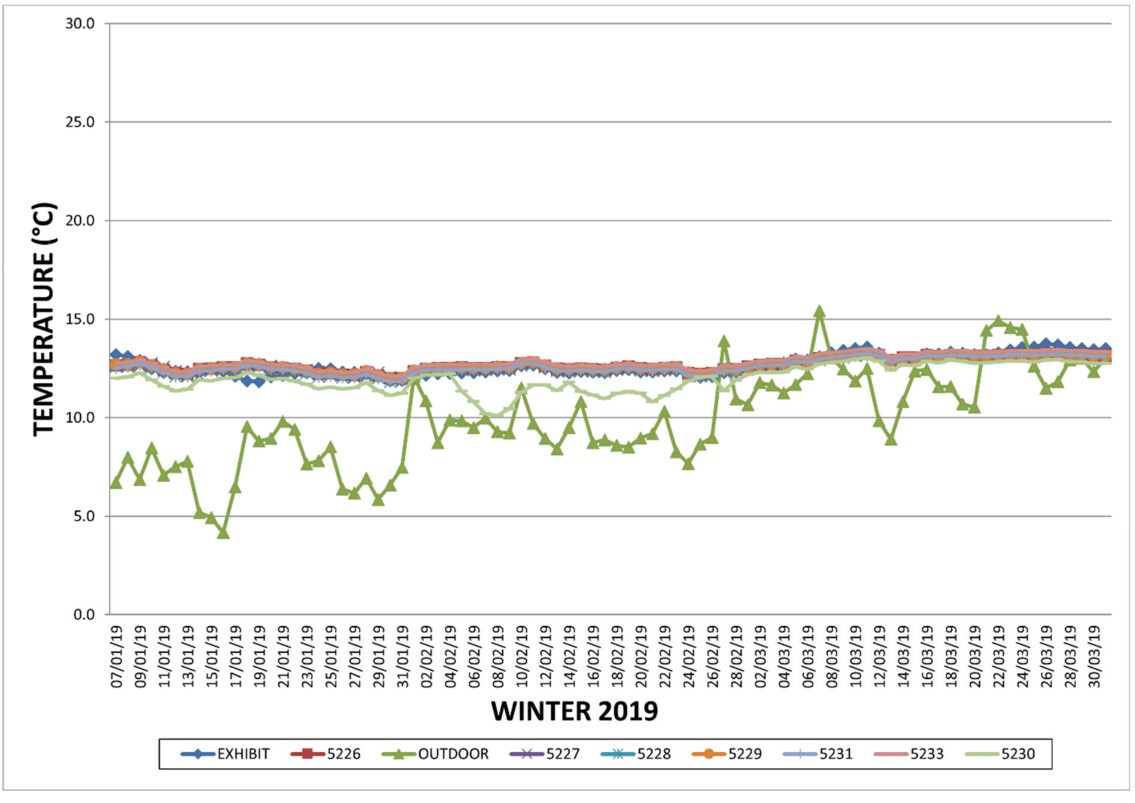

**Figure A3.** The indoor temperature values inside the exhibition area and the Mithraic gallery in comparison with the outdoor values during winter 2019 (from 1 January to 31 March).

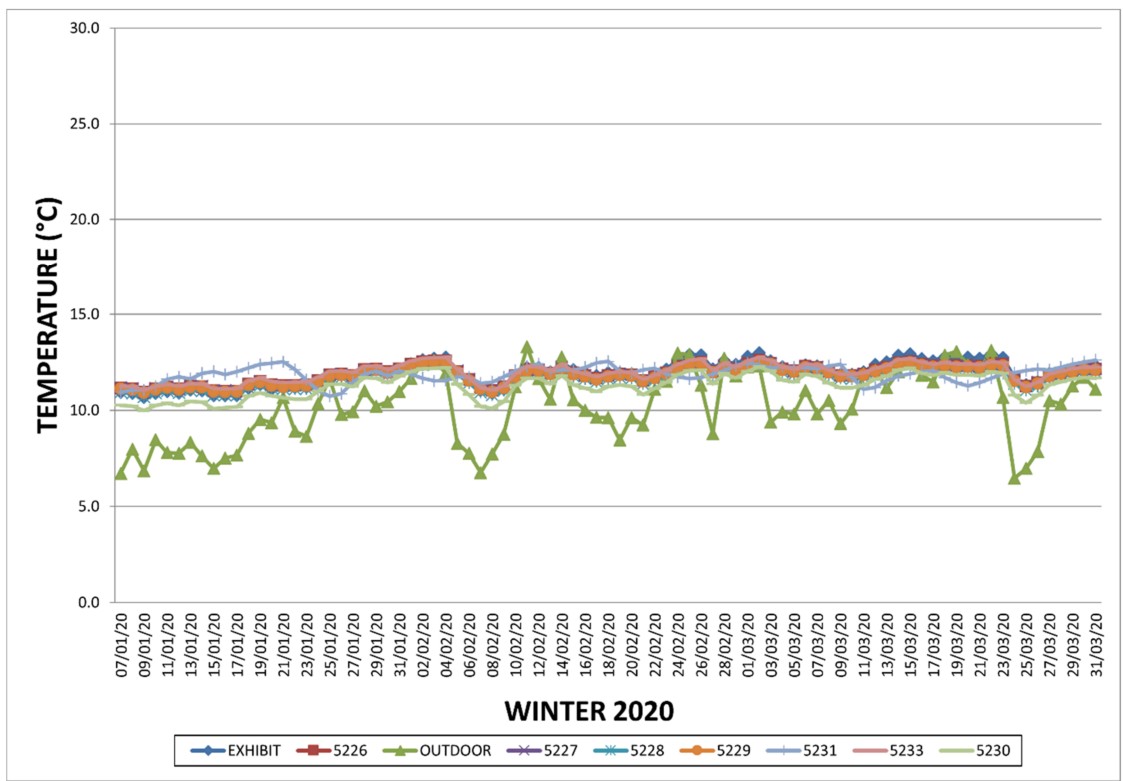

**Figure A4.** The indoor temperature values inside the exhibition area and the Mithraic gallery in comparison with the outdoor values during winter 2020 (from 1 January to 31 March).

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
