# Peer review of "Diagnostics and Monitoring to Preserve a Hypogeum Site: The Case of the Mithraeum of Marino Laziale (Rome)"

_heritage, doi:10.3390/heritage4040235_

Round 1

Reviewer 1 Report

The paper provides a holistic view on a unique heritage site and summarizes its history and past treatments, this is highly appreciated.

The analytical techniques chosen are adequate and well described. One exception is the thermal imaging. The method is described on page 8 and backed up with 6 references. However, there is no significant result included and if there was none, this needs to be better justified.

Regarding the figures, attention please: Fig 8, the figure caption is not complete. Fig 9 to 12: what is the second graph? If it is outdoor RH, this needs to be said. Attention to the labeling of the y-axis: now it is "RELATIVE UMIDITY R%" but it should be "Relative humidity RH%".

Some sentences are long and winding and can be shorter and more precise. Example from page 13: "In this area the MC is higher in percentages than the values obtained on the Mithraic scene. (delete the words in BOLD).

Some expressions appear slightly off to the reviewer:
"environmental temperature" - do you mean "air temperature"?
"contact temperature" do you mean "surface temperature"?
page 9, "download the historical time series" - do you mean "download the previous records"?

Some expressions are not correct:
page 12 "conservative treatment" - you probably mean "conservation treatment", since "conservative" is used as the opposite to "aggressive treatment";
page 18: "continuative research" should be "continuation of research".

Several sentences have no verb or are unclear, such as the first sentence of the conclusion (what is "heritage-making"?)

The quality of the paper is better in the beginning and seem to degrade towards the end. The conclusion needs to be clear and to the point and well written.

Overall, the topic is important and deserves publishing, after improving the text.

Author Response

Dear reviewer,

We would like to thank you very much for your valuable comments and suggestions, which allowed us to clarify and refine our paper.

You can find in the attached file our replies to your comments highlighted in blue.

Best regards,

Loredana Luvidi

Reviewer 2 Report

This paper presents a very interesting case study about a Mithraeum in Marino Lazaile.

I have some questions and suggestions for authors.

In the introduction, the authors explain that it is important to estimate the impact of public on the internal microclimate of hypogea sites as well as on the development of microorganisms. In the paper they do not show any study in this aspect nor comparison with other studies done in similar sites. For me this is the main lack of the paper that can be corrected by some bibliographic research about previous works in similar environments.

I would like to find the presentation of the kind of site they are investigating (chapter 2.1) in the introduction. This will help neophytes to better understand the site and its importance. 

An important amount of data has been collected and the results presented but they are not really exploited in the discussion in a global way, just one by one.

In page 13 the authors talk about counter-walls that have been removed but I did not find this information in the description of the site. May the authors, please, explain when they have been detached ?

Concerning microclimate data for inside points, it is almost impossible to visualize it on graphs. May you please use a double scale, one for outdoor data and on for indoor data ?

In conclusion this is a very interesting paper with data collected during two years and that can be a very nice baseline study for comparison when the site will be open. I would like to have some details about possible evolution based in similar sites already open to public.

I recommend its publication with some revisions.

Author Response

(The authors gave the same response as above.)

Reviewer 3 Report

The article describes an important historical site of the Mithraeum of Marino Laziale, the history of its conservation and the recently performed analytical documentation. The paper is well written, clear and informative. It could be of interest for both historians specializing in the specific period and conservators dealing with monuments. Additionally, this article can also be read by the general public.

The analytical methodology applied in this study is well described. The selection of various analytical methods is justified. The substantial part of the paper is devoted to the historical context of the case study and is described exhaustively. The paper is recommended for publication in the journal of Heritage. Only two remarks/suggestions described below might improve the paper, so the authors are invited to address them.

First - Table 1 and Methods subchapter. It is suggested to pay more attention to the description of the sampling procedure. Such aspects as: the number of spots samples (why nine and not more or less), the areas samples (why those and not others), the size and weight of samples, how they were collected, with what tools. It might also be useful to include a few images of the studies samples, e.g. microscopy images with visible and the UV light. What about the layer stratigraphy and its analysis based on the microscopy data?

Second - as a suggestion to enrich the knowledge on the state of conservation and the damages of the wall painting, it is worth analyzing the painting materials – pigments and binders – and see, if/how they were affected by the humidity/temperature/light/visitors etc. The authors seem to have some useful tools that are able to provide this information. Perhaps, it is worth returning to the collected data and extracting this information from there instead of returning to the site for more data collection.

Author Response

(The authors gave the same response as above.)
